# Whole Exome Sequencing of Thymoma Patients Exhibiting Exceptional Responses to Pemetrexed Monotherapy

**DOI:** 10.3390/cancers15164018

**Published:** 2023-08-08

**Authors:** Tomohiro Tanaka, Yasushi Goto, Masafumi Horie, Ken Masuda, Yuki Shinno, Yuji Matsumoto, Yusuke Okuma, Tatsuya Yoshida, Hidehito Horinouchi, Noriko Motoi, Yasushi Yatabe, Shunichi Watanabe, Noboru Yamamoto, Yuichiro Ohe

**Affiliations:** 1Department of Thoracic Oncology, National Cancer Center Hospital, Tokyo 104-0045, Japan; 2Department of Respiratory Medicine and Infectious Diseases, Niigata University Medical & Dental Hospital, Niigata 951-8510, Japan; 3Department of Molecular and Cellular Pathology, Kanazawa University, Kanazawa 920-8640, Japan; 4Department of Pathology, Saitama Cancer Center, Saitama 362-0806, Japan; 5Department of Pathology and Clinical Laboratory, National Cancer Center Hospital, Tokyo 104-0045, Japan; 6Department of Thoracic Surgery, National Cancer Center Hospital, Tokyo 104-0045, Japan

**Keywords:** thymoma, whole-exome sequencing, pemetrexed, copy number variations, exceptional response

## Abstract

**Simple Summary:**

Pemetrexed, a multi-target anti-folate agent, is the treatment choice for advanced or metastatic thymoma at the second or later line setting. Thymoma patients occasionally show an exceptionally durable and deep response to pemetrexed treatment. Recent studies using next-generation sequencing have identified some genomic aberrations in such patients, but the mechanism underlying their sensitivity to pemetrexed remains unclear. This study explores certain somatic single-nucleotide variants or copy number variations (CNVs) in exceptional responders to pemetrexed treatment. To elucidate any genomic changes, we performed whole-exome sequencing in patients with advanced thymomas treated with pemetrexed. We found no differences between the exceptional and typical responders, but the highest number of whole arm gain or the loss of chromosomal CNVs was observed in an exceptional responder to pemetrexed. Our study provides additional genomic findings on thymomas and chemosensitivity.

**Abstract:**

Background: Pemetrexed is used for the chemotherapy of advanced thymoma. Exceptional responses of thymoma to pemetrexed treatment are not frequently observed. The underlying genetic mechanism of the exceptional responses remains unclear. We used whole-exome sequencing to explore the specific genomic aberrations that lead to an extreme and durable response. Methods: Whole-exome sequencing using NovaSeq6000 (150 bp paired-end sequencing) was performed on nine formalin-fixed paraffin-embedded tissues from patients with advanced thymomas treated with pemetrexed (two exceptional responders and seven typical responders). Results: We identified 284 somatic single-nucleotide variants (SNVs; 272 missense, 8 missense/splice-site, 3 stop-gain, and 1 stop-gain/splice-site), 34 insertions and deletions (Indels; 33 frameshift and one splice region), and 21 copy number variations (CNVs; 15 gains and six losses). No difference in the number of SNVs variants and distribution of deleterious Indels was observed between the exceptional and typical responders. Interestingly, arm-level chromosomal CNVs (15 gains and six losses) were detected in four patients, including an exceptional responder. The highest number of arm-level CNVs was observed in an exceptional responder. Conclusion: Exceptional responders to pemetrexed for metastatic thymomas may be characterized by arm-level CNVs. Further, whole-genome and RNA sequencing studies should be performed.

## 1. Introduction

Thymomas are rare anterior mediastinal neoplasms that arise from thymic epithelial cells and are classified according to the World Health Organization (WHO) by histopathological classification (subtypes A, AB, B1, B2, B3, and carcinoma) [1]. Surgical resection cures early-stage thymomas, whereas systemic chemotherapy is required for metastatic or recurrent thymomas. Although anthracycline-based chemotherapy is commonly used as first-line treatment for advanced thymomas [2,3]; however, no standard chemotherapy regimen has been established.

Pemetrexed is an approved multi-target anti-folate agent for the treatment of malignant pleural mesothelioma and non-small cell lung carcinoma [4]. A single-arm phase II trial [5] showed that pemetrexed is also useful for the treatment of heavily pretreated thymomas. Pemetrexed rarely produces an exceptional response in advanced thymoma, i.e., a profound and durable reduction in tumor size. In a small subset of patients with solid tumors, molecular-targeted treatments lead to extreme responses [6,7,8,9,10]. In previous studies, genomic alterations on the mammalian target of rapamycin (mTOR) pathway activation conferred exceptional response to everolimus or pazopanib [7] in various tumors [6]. Additionally, patients with lung adenocarcinoma exhibiting *KRAS* mutation and *EGFR* amplification showed an exceptional response to erlotinib therapy [8]. Another study identified very few extreme responders to anticancer therapy among patients with advanced breast cancer [9]. Moreover, patients with low-grade ovarian cancer showed an exceptional clinical response to ibrutinib when guided by organoid drug testing [10]. In these cases, certain genomic aberrations may confer exceptional sensitivity to target therapies [7,9].

Recently, next-generation sequencing (NGS) studies have revealed new genomic mutations in several tumors, including the lung, breast, and colon [11,12,13]. Additionally, case studies that performed NGS found genomic aberrations in thymomas, such as *GTF2I* [14,15], *ASXL2*, *DNMT3A* [16], *KRAS*, *HRAS* [17], *CDKN2A/B* [18], and *TP53* [18], as well as chromosomal abnormalities [19,20]. However, the molecular basis of pemetrexed sensitivity in thymomas remains unclear [21,22].

However, to date, no predictive markers for pemetrexed treatment are available. Previous studies reported that mRNA levels of *TPX2*, *CPA3*, *EZH2*, *MCM2,* and *TOP2A* [23], a folylpoly-γ-glutamate synthase single nucleotide polymorphism [24], may be potential predictive biomarkers of pemetrexed sensitivity in non-small cell lung cancer. On the other hand, elevated thymidylate synthase expression levels, including [25] *BMI1* [26], *AMPK* [27], and *ABCC5* [28], may confer acquired resistance to pemetrexed or predict poor therapeutic sensitivity of pemetrexed chemotherapy. The useful predictive molecular markers for pemetrexed-based treatment have been expected.

Furthermore, genomic mutations associated with oncogenic drivers, particularly drug sensitivity, have not been identified in thymomas by targeted sequencing.

In the present study, we performed the whole-exome sequencing of patients with thymoma, including those who showed an exceptional response to pemetrexed treatment. The purpose of this study was to identify genomic alterations that could predict an exceptional treatment response to pemetrexed.

## 2. Materials and Methods

### 2.1. Patients and Tissue Collection

We retrospectively reviewed the medical records of 12 Japanese patients with advanced thymoma treated with pemetrexed monotherapy between August 2011 and April 2020 at the Department of Thoracic Oncology, National Cancer Center Hospital, Tokyo, Japan. Formalin-fixed paraffin-embedded (FFPE) thymoma samples were collected from six hospitals in Japan. The histopathological analysis was performed by expert pathologists in accordance with the 2015 WHO classification.

### 2.2. Assessment and Tumor Response

Pemetrexed was administered at 500 mg/m^2^ every 21 days. Clinical efficacy was assessed every 2–3 months. The tumor response was evaluated using The Response Criteria in Solid Tumors (version 1.1) [29]. The objective response rate (ORR) was defined as the proportion of patients who achieved a complete response (CR) or partial response (PR). The disease control rate (DCR) was defined as proportion of the patients who achieved CR, PR, or stable disease.

### 2.3. Definition of Exceptional Response to Pemetrexed

Based on previous studies [7,9], a CR or PR to pemetrexed monotherapy for 15 months until disease progression or death (equivalent to progression-free survival [PFS]) was considered to indicate an exceptional response.

### 2.4. DNA Extraction

Genomic DNA was extracted from FFPE specimens from advanced thymomas using the Maxwell RSC DNA FFPE Kit (Promega, Madison, WI, USA). A DNA library was prepared using the SureSelect Human All Exon Kit v6 (Agilent Technologies Inc., Santa Clara, CA, USA) in accordance with the manufacturer’s guidelines. The DNA quality and quantity were evaluated using a 2100 Bioanalyzer DNA 1000 kit (Agilent Technologies Inc.). The total DNA (0.2 µg) met the quality control criteria and was determined to be acceptable for analysis. Genomic DNA was fragmented, purified, end-repaired, adenylated on the 3′ ends, ligated to indexed pair-end adaptors, purified again, and amplified by PCR. The genomic DNA was then cleaved into fragments ranging from approximately 236 to 367 bp, and sequencing adapters were attached to these fragments. We subsequently performed whole-exome sequencing using NovaSeq6000 (150 bp paired-end sequencing) (Illumina, San Diego, CA, USA).

### 2.5. Sequencing Analysis

The obtained FASTQ files were aligned to the hg38 human reference using Burrows–Wheeler Aligner (version 0.7.17) (Appendix A). The output binary alignment map files were then processed using the MarkDuplicates tool in Picard (version 2.18.2). For somatic analysis, the base-recalibrated tumors were analyzed using GATK (version 4.0.5.1) to call somatic variants at chromosome positions covered in the target bed (Agilent SureSelect V6). The resulting somatic variant call files were hard-filtered at a read depth of ≥20. Realignment and somatic variant calling were performed using Mutect2. Genetic single-nucleotide variants (SNVs) and insertions and deletions (Indels) were annotated using SnpEff (version 4.3).

To retain high-confidence nonsynonymous coding variants, we applied the following criteria for the additional filtration of the initial call set: (1) filter depth > 20, (2) quality score > 100, and (3) the removal of common germline single-nucleotide polymorphisms using a public database (dbSNP 138). Detected SNVs were searched using information from publicly available databases, including the Genome Aggregation Database, 1000G-integration, ExAC, and ExAC non-TCGA, to obtain a <0.01 minor allele frequency.

SNVs were identified as potentially pathogenic if they were identified by at least two of the following methods: SIFT, LRT, MutationTaster, MutationAssessor, FATHMM, PROVEAN, MetaSVM, MetaLR, MCAP, and fathmm-MKL (coding).

### 2.6. Copy Number Variation (CNV) Analysis

Briefly, the CNVs were analyzed, and heatmap plots were drawn using CNVkit (version 0.9.6) [30]. Data were entered into the CNVkit, and copy number calls were generated for each sample based on the cutoff value for deletions and duplications specified in the settings. The identified CNVs were interpreted according to standard methods, and the data were normalized according to the average depth. The median CNV was used as the cutoff for classifying CNVs in our cohort. The gene copy number gain and loss were indicated by all exons that had a copy number > 0.3 and <−0.3 log2, respectively.

### 2.7. Statistical Analysis

Statistical analysis was performed using the EZR program (version 1.37; Saitama Medical Center, Jichi Medical University, Saitama, Japan) [31]. Overall survival (OS) curves were drawn using the Kaplan–Meier method. *p* < 0.05 was considered statistically significant. *p* values and hazard ratios were calculated using the log-rank test.

## 3. Results

### 3.1. Patient Characteristics and Sample Collection

Figure 1 shows the CONSORT diagram. We identified 38 patients with advanced thymoma who visited the participating institutes; of these patients, 12 (three exceptional responders and nine typical responders) were included in the study. One exceptional responder and one typical responder were excluded from the analysis because they had no or insufficient tumor specimens available for sequencing. Therefore, the final analysis included two exceptional responders (patients 3 and 8) and seven typical responders (Figure 2). Of the nine thymoma patients, one (11%), two (22%), four (44%), and two (22%) had B1, B2, B3, and B2 + B3 WHO subtype thymomas, respectively (Table 1). The patients included two males and seven females, with a median age of 51 years (range: 36–77 years). Three (33%) patients met the stage III Masaoka criteria at diagnosis, whereas six (66%) met the stage IV criteria.

All patients had an Eastern Cooperative Oncology Group performance status of 0–1. Of the responders, 20 (67%) had received pemetrexed as a first-line treatment, whereas 7 (78%) had previously received anthracycline-containing regimens, 1 (11%) had received paclitaxel, and 1 (11%) was chemotherapy-naïve.

### 3.2. Response and Survival Analysis

After a median follow-up period of 18.0 months (95% confidence interval [CI] = 12.6–29.1), six patients experienced recurrence, and two died. The patients underwent a median of 10 (range: 2–33) pemetrexed treatment cycles. Of the nine patients, PR and stable disease were achieved in two and five, respectively, for a response rate of 22.2% (95% CI = 3.8–60.0) and a DCR of 77.8% (95% CI = 21.2–86.3). The median PFS of all patients was 16.0 months [95% CI = 1.0 − not reached (NR)]. The median OS was NR (95% CI = 12 − NR). The median PFS of exceptional responders was longer than that of typical responders (22.5 vs. 3.1 months; *p* = 0.37) (Figure 3A,B).

### 3.3. Whole-Exome Sequencing

Whole-exome sequencing was performed on the nine FFPE samples from the nine thymoma patients. We used an analysis pipeline that involved rigorous quality control (QC) and filtering. To identify the mutations, we obtained an average sequencing at a depth of 153.4× (range: 39.7–205×) in tumor DNA. Whole-exome sequencing was not performed for patient 7 because of low coverage depth.

### 3.4. Identification of SNVs and Indels

In total, 284 somatic SNVs were identified in the tumor genomes (Appendix A). A median of 35.1 variants per sample was identified in the samples (Figure 4A). There was no difference in variants between exceptional and typical responders.

With regard to transversions, G>A substitutions were the most common mutation in all samples (50/271, 18.5%), indicating formalin fixation artifacts (Figure 4B). The second most common mutation was C>T substitution (41/271, 15.1%). The mutation pattern was similar between exceptional and typical responders. Furthermore, 17 SNVs were identified in genes that were predicted to be potentially pathogenic by at least one of the above-mentioned methods (Table 2).

Next, we filtered out the 282 Indels. The remaining 34 mutations included frameshift mutations (Table 3). Of the Indels, variations in *GTF3C1*, *SCN3A*, and *RPL5* were identified as potentially deleterious if they were detected more than twice using computational filtered methods. No difference in the distribution of the deleterious Indels was identified between exceptional and typical responders to pemetrexed. Unlike previous studies, aberrations were not detected in *GTF2I*, *ASXL2*, *DNMT3A*, and *KRAS*/*HRAS*.

### 3.5. Identification of CNVs

Of the nine samples, two were excluded (patients 3 and 7) from the CNV analysis to reduce the number of false positives arising due to a low gene coverage depth (average read depth < 100×). Consequently, data from one exceptional responder and six typical responders were analyzed.

According to previous studies, CNVs that covered > 25% [32] to 33% of the chromosome arm [32] were considered arm-level CNVs; these were detected in 57.1% (4/7) of thymoma patients (with a total of 15 copy number gains and six copy number losses) (Table 4 and Appendix A, Appendix A).

The most common chromosomal gain was 1q (4/7, 57.1%), followed by 5p (3/7, 42.9%), 9p (2/7, 28.6%), and 3p (1/7, 14.3%) (Table 5). The most common whole-arm gain was chromosome 7 (1/7, 14.3%), followed by chromosome 14 (1/7, 14.3%). The most common chromosomal loss was the whole arm of chromosome 6 (2/7, 28.6%), followed by 2p (1/7, 14.3%) and 3p (1/7, 14.3%). The whole-arm chromosome gain was seen most commonly on chromosomes 7 (2/7, 28.6%), 5 (1/7, 14.3%), and 14 (1/7, 14.3%). Whole-arm chromosome loss was seen most commonly on chromosomes 6 (2/7, 28/6%) and 13 (1/7, 14.3%).

A hierarchical clustering algorithm generated four distinct clusters (Figure 5).

Cluster 1 (patients 4 and 8) was characterized by multiple chromosomal arm-level CNVs, whereas cluster 3 (patients 6 and 9) showed arm-level chromosome gain. An exceptional responder had the most arm-level CNVs (patient 8).

## 4. Discussion

In this study, we performed the whole-exome sequencing of tissues from patients with advanced thymoma to explore the genomic variants related to exceptional responses of pemetrexed monotherapy. We identified certain Indels that were potentially deleterious, including *SCN3A*, *GTF3C1*, and *RPL5*. However, none of the genomic alterations were candidate driver oncogenes, and none were specific to exceptional responders. No novel driver oncogene was identified in advanced thymomas. Interestingly, arm-level chromosomal CNVs were detected in some patients with advanced thymomas treated with pemetrexed.

Cluster analysis revealed that cluster 1, which was detected in an exceptional responder (patient 8), was associated with remarkably increased arm-level chromosomal CNVs, suggesting that arm-level chromosomal CNVs might be related to the response to pemetrexed. A previous study showed that the *MYC* chromosomal copy number was related to the clinical response to paclitaxel and the in vitro antitumor effect of mTORC1/2 inhibition [33]. Another retrospective study showed that chromosomal 1q gain may have a detrimental impact on the prognosis of multiple myeloma when treated with bortezomib-based regimens [34]. However, that study included only one exceptional responder.

Our results also suggest that arm-level chromosomal CNVs are present mainly in type B2 or B3 thymomas with a chromosomal gain of 1q, 5p, 9p, or 3p; with chromosomal loss of 2p or 3p; with a whole-arm chromosomal gain of 7, 5, or 14; or with whole-arm chromosomal loss of 6 or 13. Previous studies have found that chromosomal arm-level CNVs were detectable mainly in the thymoma subtypes B2 and B3 (Appendix A) [17,20,35,36]. These studies showed that some type B3 thymomas were characterized by chromosomal 1q gain and chromosomal 6 loss [17,20]. Another study revealed the loss of 13q in type B3 thymomas [20]. Petrini et al. [35] reported a case of B3 thymoma with the gain of chromosomes 1q, 5, and 7 and the loss of chromosomes 3p, 6, 13, and part of chromosome 11q. A comprehensive analysis of The Cancer Genome Atlas (TCGA) showed that CNVs are rare in type A and AB thymomas, whereas arm-level CNVs are frequently detected in type B2 and B3 thymomas [37]. These results suggest that chromosomal arm-level CNVs might be related to the aggressiveness of thymomas. These results suggest that chromosomal arm-level CNVs might be related to the aggressiveness of thymomas. A recent study showed that arm-level CNVs can be used to screen early-stage colorectal cancer [38] or lung cancer [39]. However, the clinical significance of arm-level CNVs as a screening tool for early thymomas remains unclear. Although arm-level CNVs might play a role in thymoma oncogenesis, this is difficult to confirm because of the rarity of thymomas.

In this study, chromosomal 1q gain and whole chromosome 6 loss were detected in advanced thymomas. In other tumors, chromosome 1q gain was identified in multiple myeloma [34,40], ependymoma [41], Wilms tumor [42], and New Zealand virus-negative Merkel cell carcinomas [43]. Whole chromosome 6 loss has been detected in acute lymphoblastic lymphoma [44]. The gain of whole chromosome 9, the loss of whole chromosome 22, and the loss of the Y chromosome were detected in polymorphous low-grade adenocarcinoma of the head and neck [45]. The loss of whole chromosome 7 or loss of 7q and 5q were the most frequent primary abnormalities significantly related to myelodysplasia [46].

With respect to the prognosis or malignant grade, chromosomal 1q gain predict a poor prognosis of hepatocellular carcinoma [47]. Additionally, the aggressiveness of diffuse leptomeningeal glioneuronal tumors might be related to chromosomal 1q gain [48]. Another study showed that the loss of whole chromosome 3, the loss of 6q, and the gain of 8q were significantly associated with poor overall survival in patients with metastatic primary uveal melanoma [49]. The co-occurrence of chromosomal 1p loss and 1q gain was associated with a poor prognosis in patients with low-grade serous ovarian carcinoma [50]. The gain of whole chromosome 7 showed a more aggressive clinical nature in grade II and III gliomas [51]. However, no study to date has clarified whether chromosomal gain or loss has any effect on the prognosis or malignancy of thymomas.

TCGA suggested that patients with thymoma combined with myasthenia gravis (MG) have more chromosomal arm-level CNVs [37]. Another study suggested that the loss of whole chromosome 6, including the HLA locus, might play a role in paraneoplastic autoimmunity in patients with type B3 thymoma [20]. In our study, only patient 2 had concomitant MG; this patient had chromosomal arm-level CNVs. However, the sample size of the present study was insufficient to evaluate the relationship between MG and arm-level CNVs.

Certain potentially harmful SNVs were identified in the present study. Among these, a *STAT6* missense variant was reported as a potential cause of primary atopic disorders [52]. Additionally, the *ERBB2* missense mutations D769Y and D742N were associated with acquired resistance to tyrosine kinase inhibitors [53]. The missense V419L variant in *TGFBR2* associated with Loeys–Dietz syndrome was also related to impaired TGF-β signaling [54]. Patients with pathogenic missense mutations in *LZTR1* exhibited a characteristic Noonan syndrome [55]. A missense mutation in *NROB1* was associated with isolated mineralocorticoid deficiency [56]. A missense mutation in *HERC2* was associated with intellectual disability, autism, and gait disturbance [57]. *PCDHA9* has been identified as a potential candidate gene for Hirschsprung’s disease [58]. A homozygous missense mutation in *STIL* causes holoprosencephaly and microcephaly [59]. In one study, genetic aberrations in *FBXW7* were detected in 55% of patients with primary uveal melanoma [60]. A recent study showed that *EPHA5* mutations, along with other molecular alterations in the DNA damage response pathway and a favorable antitumor immune signature, could contribute to exceptional responses [61]. These deleterious SNVs do not seem to be related to the oncogenesis of thymoma or sensitivity to pemetrexed.

This study identified the potentially deleterious Indels *SCN3A*, *GTF3C1*, and *RPL5*. These mutations might have played oncogenic roles in the advanced thymomas in our study. A previous study showed that *SCN3A*, which encodes the voltage-gated sodium channel subunit Nav1.3, caused severe epilepsy and disordered cortical development [62]. *GTF3C1* is part of the *GTF3* family, which is related to the expansion of different types of cancers [63]. 1p22 deletion encompassing the *RPL5* gene caused Diamond–Blackfan anemia [64]. However, the same deleterious Indels were not observed in other cases.

Furthermore, these deleterious SNVs and Indels were not detected in patients with advanced thymomas or exceptional sensitivity to pemetrexed. Although certain SNVs or Indels appeared to be harmful, no deleterious SNVs or Indels were found to be candidates for oncogenic driver mutations in relation to drug sensitivity and pemetrexed in patients with advanced thymomas.

Regarding pemetrexed, two previous studies performed the whole-exome sequencing of non-small cell carcinoma patients with exceptional responses to pemetrexed and carboplatin [61,65]. However, the molecular aberrations identified in the previous studies did not match those identified in the present one. Therefore, the underlying genetic mechanism of exceptional response to pemetrexed might differ between non-small cell carcinoma and thymomas. In particular, an exceptional responder (patient 8) had a whole-arm gain and loss of chromosome 3 that contained the *CAND2* genes. *CAND2* is a translationally upregulated gene that encodes a muscle-specific protein and is dependent on the activity of mTOR [66]. *CAND2* has also been identified as a novel obesity susceptibility gene [67]. In the present study, we were unable to identify a relationship between *CAND2* and its sensitivity to pemetrexed. Furthermore, this was seen in only one patient. Further studies are required to validate our results. *GTF2I* mutations, which were frequently found in the comprehensive genomic profiling study of thymomas [14], were not detected in the present study. *GTF2I* mutations were exclusively detected in early-stage thymoma [14,37]. However, our study did not include patients with thymoma subtypes A or AB.

Our study had some limitations. First, the number and size of the tumor samples were limited, which prevented us from drawing definitive conclusions. In particular, only two exceptional responders were included, one of whom was excluded from the analysis of CNVs due to a small amount of genomic data. No further conclusions could be drawn from this single case. Second, sequencing is difficult to perform on archived FFPE samples, and our RNA sequencing, including enrichment analysis using the Gene Ontology/Kyoto Encyclopedia of Genes and Genomes analysis of the CNVs, failed because the RNA of almost all samples was fragmented. Therefore, the fusion genes of thymomas could not be explored. Future studies should collect frozen tissues of advanced thymoma patients, including exceptional responders, to allow for comprehensive analysis of genomic aberrations, including mutations, CNVs, and structural genomic variations by whole-genome or RNA sequencing.

In conclusion, this is the first study to perform the whole-exome sequencing of patients with advanced thymomas treated with pemetrexed. Although we failed to identify the molecular mechanisms of exceptional response to pemetrexed, arm-level CNVs could be related to the genetic mechanism underlying extreme sensitivity to pemetrexed treatment or the deterioration of the thymoma. Further studies are required to confirm our results.

## 5. Conclusions

In the present study, none of the evaluated genomic alterations were specific to exceptional responders to pemetrexed treatment, and no new driver gene was identified in advanced thymomas. However, arm-level chromosomal CNVs were detected in four of seven patients. Because this was a retrospective study of a small sample, the results will not affect the treatment choice for thymomas. Further, whole-genome and RNA sequencing studies should be performed.

## Figures and Tables

**Figure 1 cancers-15-04018-f001:**
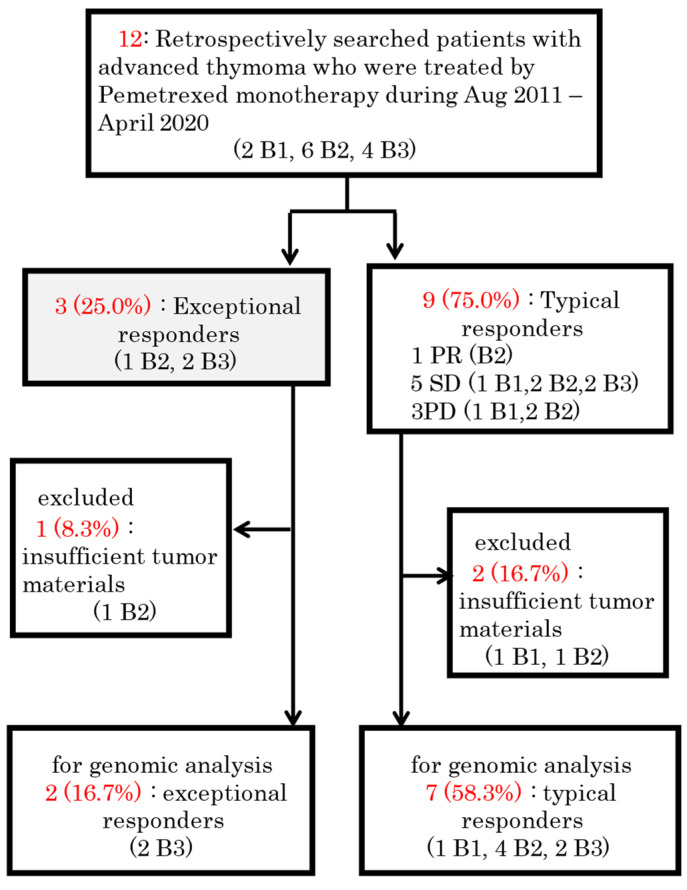
Schematic diagram of the study protocol. PR = partial response. SD = stable disease. PD = progressive disease.

**Figure 2 cancers-15-04018-f002:**
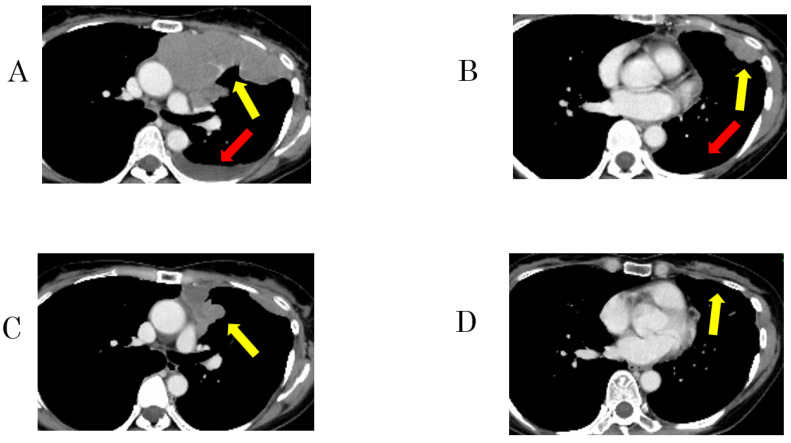
Representative imaging of a 51-year-old woman with type B3 thymoma showing an exceptional response to pemetrexed monotherapy (patient 8). (**A**,**B**) Enhanced computed tomography (CT) revealed massive pleural dissemination and pleural effusion in the left hemithorax before pemetrexed treatment. (**C**,**D**) After 7 months of pemetrexed treatment, CT showed shrinkage of the massive lesions. Images of pleural dissemination (yellow arrows) and pleural effusion (red arrows).

**Figure 3 cancers-15-04018-f003:**
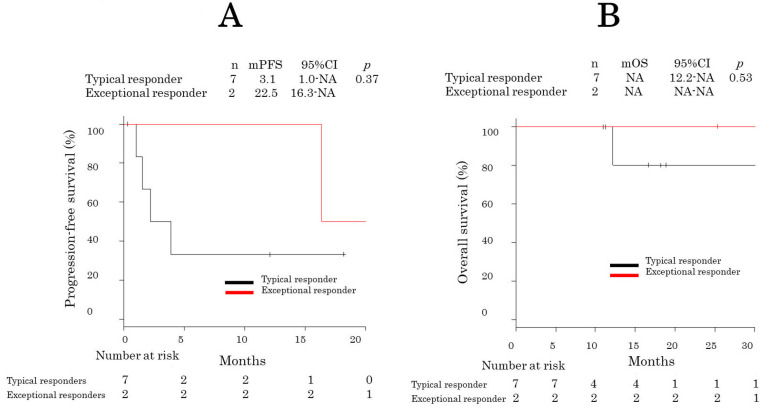
Kaplan–Meier progression-free survival curves of exceptional and typical responders. Kaplan–Meier progression-free survival (**A**) and overall survival (**B**) curves.

**Figure 4 cancers-15-04018-f004:**
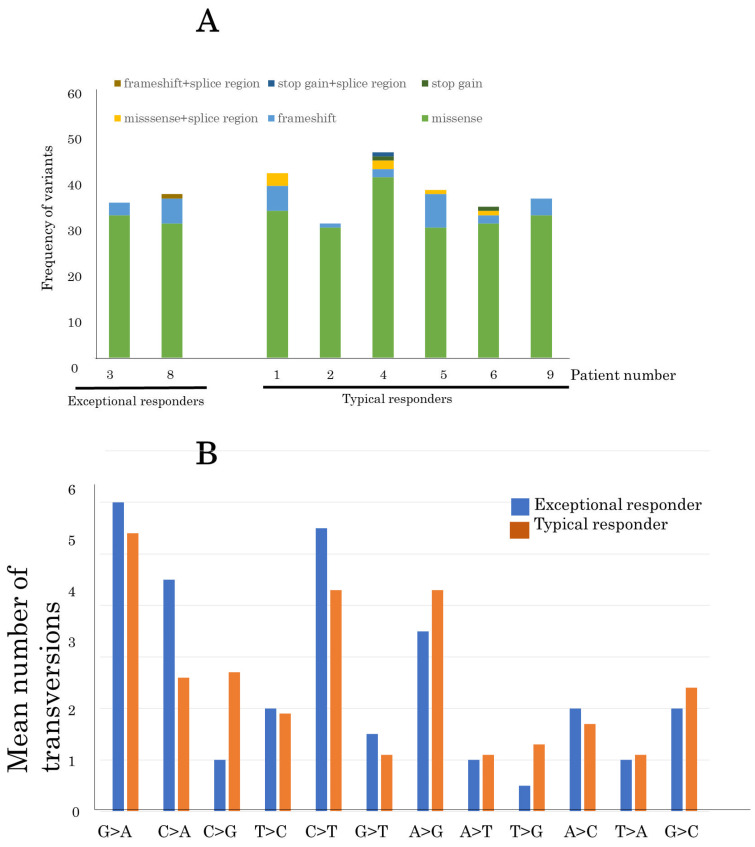
Comparison of single-nucleotide variants between exceptional and typical responders (**A**) Comparison of the frequency of variants in advanced thymoma patients between exceptional and typical responders to pemetrexed. (**B**) The mean number of somatic transversion mutations in nine advanced thymomas from exceptional and typical responders.

**Figure 5 cancers-15-04018-f005:**
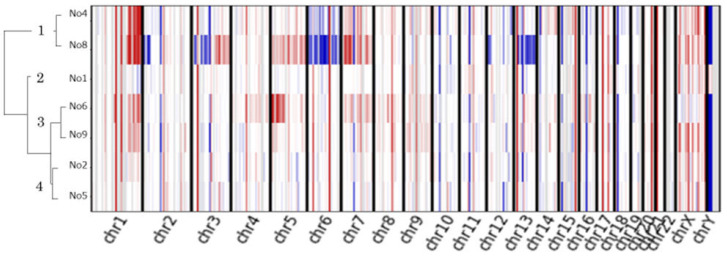
Copy number analysis of nine advanced thymomas. Unsupervised hierarchical clustering heatmap for the nine thymoma patients treated with pemetrexed; copy number gains are depicted in red, and losses are in blue.

**Table 1 cancers-15-04018-t001:** Clinical characteristics of 9 Thymoma patients treated by pemetrexed monotherapy.

No.	Gender	Age	WHO-Subtypes	Masaoka-Stage	PS	Line of Therapy	Previous Chemotherapy	Paraneoplastic Syndrome	Response	Eceptional Response	Mean Depth of Target Lesion (×)	PFS, Months	OS, Months
1	M	61	B1	IVa	1	2	CBDCA+PTX	non	SD	no	161.7	3.9	12.1
2	F	43	B3	IVa	1	2	ADOC	non	SD	no	179.3	NR	NR
3	F	51	B3	III	1	3	CBDCA+PTX, CODE	MG	PR	yes	96.5	28.6	NR
4	F	66	B3	IVb	0	2	ADOC	non	SD	no	147.8	NR	NR
5	F	54	B2+B3	III	1	2	CODE	PRCA	SD	no	177.6	NR	NR
6	F	36	B2+B3	IVb	1	2	CODE	non	PD	no	189.1	2.2	NR
7	F	77	B2	IVa	1	3	AMR, CODE	non	SD	no	39.7	1.0	NR
8	F	51	B3	IVa	1	2	CODE	MG	PR	yes	205.2	16.3	NR
9	F	45	B2	III	0	1	non	non	PD	no	183.7	1.5	NR

Abbreviations: No.: number. WHO: World Health Organization. PS: performance status. PFS: progression free survival. OS: overall survival. CBDCA: carboplatin. PTX: paclitaxel. ADOC: adriamycin + cisplatin + vincristine + cyclophosphamide. CODE: cisplatin + vincristine + doxorubicin + etoposide. AMR: amurubicin. MG: myathenia gravis. PRCA: pure red cell aplasia. SD: stable disease. PR: partial response. PD: progressive disease. NR: not reached.

**Table 2 cancers-15-04018-t002:** List of SNVs that were predicted to be deletorious.

Patient No	Gene Name	Position	Nucleotide	AF	Effect
1	*STAT6*	chr12:57,102,481	c.1321G>A	0.3783784	missense
1	*CFAP54*	chr12: 96,489,820	c.211C>T	0.4841629	missense
1	*ERBBB2*	chr17:39,723,566	c.2114C>T	0.3850932	missense
1	*BIRC6*	chr2:32,431,088	c.3246C>A	0.50	missense and splice region
1	*TGFBR2*	chr3:30,623,201	c.97G>A	0.3109244	missense and splice region
2	*SACS*	chr13:23,336,092	c.7784C>T	0.4642857	missense
2	*UBR5*	chr8:102,286,377	c.5198G>A	0.4926829	missense
3	*HNF1A*	chr12:120,993,565	c.572G>A	0.6333333	missense
3	*LZTR1*	chr22:20,988,851	c.572C>T	0.5785441	missense
4	*TDRD6*	chr6:46,688,924	c.796C>A	0.3931624	missense
4	*UTRN*	chr6:144,782,110	c.8821A>G	0.2698413	missense
4	*PKHD1L1*	chr8:109,401,516	c.1301C>A	0.4382022	missense
5	*PAM*	chr5:102,974,297	c.1344G>C	0.4698795	missense
5	*NROB1*	chrX:30,308,879	c.485C>T	0.4477825	missense
6	*DOCK2*	chr5:169,759,723	c.2395A>C	0.75	missense
6	*ANK1*	chr8:169,759,723	c.2473G>A	0.3671875	missense
8	*HERC2*	chr15:28,255,910	c.2833G>A	0.4848485	missense
8	*ATP2C2*	chr16:84,422,514	c.749C>T	0.4411765	missense
8	*AXIN2*	chr17:65,549,596	c.880G>C	0.5213033	missense
8	*PCDHA9*	chr5:140,850,275	c.1780G>A	0.6423611	missense
9	*STIL*	chr1:47,280,717	c.1741T>G	0.6149068	missense
9	*ZFXH2*	chr16:72,796,380	c.6302T>C	0.3972603	missense
9	*LYL1*	chr19:13,099,622	c.540C>G	0.4915254	missense
9	*FBXW7*	chr4:152,326,028	c.1622C>G	0.4647887	missense

Abbreviations: SNVs: single nucleotide variations. AF: Allele frequency. chr: chromosome.

**Table 3 cancers-15-04018-t003:** Validated Indels detected in patients with advanced thymoma treated by Pemetrexed.

Patient No	CHROM	POS	REF	ALT	AF	Effect	Gene_Name	Deletorius	HGVS.p
1	chr11	44,919,272	C	CCTGG	0.4245077	frameshift_variant	*TSPAN18*		p.Asp132fs
1	chr16	27,470,158	C	CTT	0.5069124	frameshift_variant	*GTF3C1*	yes	p.Gln1589fs
1	chr17	50,528,794	TCA	T	0.16	frameshift_variant	*MYCBPAP*		p.Lys880fs
1	chr3	157,437,716	AG	A	0.5151515	frameshift_variant	*PTX3*		p.Arg112fs
1	chr7	34,658,494	GA	G	0.4679803	frameshift_variant	*NPSR1*		p.Thr29fs
1	chr8	141,163,405	A	AG	0.4418605	frameshift_variant	*DENND3*		p.Asp397fs
2	chr22	23,838,973	TC	T	0.52	frameshift_variant	*DERL3*		p.Gly5fs
3	chr22	23,838,973	TC	T	0.4519231	frameshift_variant	*DERL3*		p.Gly5fs
3	chr3	196,707,700	TGG	T	0.15	frameshift_variant	*CEP19*		p.Ala118fs
3	chr6	117,365,113	GTA	G	0.1025641	frameshift_variant	*ROS1*		p.Thr1022fs
4	chr12	113,163,221	CCT	C	0.5619835	frameshift_variant	*DDX54*		p.Arg664fs
4	chr12	121,954,299	GA	G	0.4193548	frameshift_variant	*WDR66*		p.Glu502fs
5	chr10	99,930,218	C	CTATATATA	0.425	frameshift_variant	*DNMBP*		p.Glu182fs
5	chr12	13,090,671	C	CT	0.4492188	frameshift_variant	*GSG1*		p.Gly66fs
5	chr16	29,797,007	A	ATCCC	0.4829396	frameshift_variant	*KIF22*		p.Pro64fs
5	chr17	41,084,543	A	AT	0.130137	frameshift_variant	*KRTAP4-7*		p.Ser113fs
5	chr17	46,549,134	ATC	A	0.2313725	frameshift_variant	*LRRC37A2*		p.Leu1333fs
5	chr2	29,920,063	G	GC	0.4986702	frameshift_variant	*ALK*		p.Arg200fs
5	chr2	165,127,882	A	AATAAT	0.4411765	frameshift_variant	*SCN3A*	yes	p.Cys1048fs
5	chr9	137,232,083	T	TC	0.4404762	frameshift_variant	*SLC34A3*		p.Ala34fs
6	chr11	68,033,138	T	TC	0.4482759	frameshift_variant	*NDUFS8*		p.Arg77fs
6	chr7	2,250,932	CG	C	0.4650206	frameshift_variant	*NUDT1*		p.Tyr159fs
8	chr1	92,833,642	CAG	C	0.3971631	frameshift_variant	*RPL5*	yes	p.Asp59fs
8	chr12	55,470,067	TACAATCTGTA	T	0.4090909	frameshift_variant	*OR6C70*		p.Leu21fs
8	chr15	79,456,702	GAAAA	G	0.1346154	frameshift_variant	*KIAA1024*		p.Lys186fs
8	chr15	79,456,706	A	ACCCC	0.1390728	frameshift_variant	*KIAA1024*		p.Asn187fs
8	chr19	43,613,237	C	CAAAG	0.4780876	frameshift_variant	*SRRM5*		p.Asp376fs
8	chr19	54,574,302	GC	G	0.1104294	frameshift_variant&	*LILRA2*		p.His25fs
						splice_region_variant			
8	chr3	12,815,222	TC	T	0.88	frameshift_variant	*CAND2*		p.Ser364fs
9	chr1	53,082,128	A	AG	0.4193548	frameshift_variant	*PODN*		p.Glu652fs
9	chr11	66,366,223	C	CAGCCAGATCTGGG	0.3613861	frameshift_variant	*SLC29A2*		p.Thr293fs
9	chr19	53,807,589	C	CT	0.4126214	frameshift_variant	*NLRP12*		p.Glu718fs
9	chr7	100,578,215	GC	G	0.5545455	frameshift_variant	*LRCH4*		p.Leu298fs

Abbreviations: Indels = insertion and deletions, CHR = chromosome, POS = position, REF = reference allele, ALT = alternative allele, AF = allele frequency, HGVS.p = human genome variation society, protein.

**Table 4 cancers-15-04018-t004:** The distribution of cromosomal arm-level CNVs per patient with advanced thymoma treated by Pemetrexed.

CHR	Patient 1	Patient 2	Patient 4	Patient 5	Patient 6	Patient 8	Patient 9
1p							
1q			gain		gain	gain	gain
2p						loss	
2q							
3p						loss	
3q						gain	
4p							
4q							
5p					gain	gain	gain
5q						gain	
6p			loss			loss	
6q			loss			loss	
7p					gain	gain	
7q					gain	gain	
8p							
8q							
9p							gain
9q							
10p							
10q							
11p							
11q							
12p							
12q							
13p						loss	
13q						loss	
14p			gain				
14q			gain				
15p							
15q							
16p							
16q							
17p							
17q							
18p							
18q							
19p							
19q							
20p							
20q							
21p							
21q							
22p							
22q							

Abbreviations: CNVs = copy number variations, CHR = c hromosome.

**Table 5 cancers-15-04018-t005:** Frequency of CNVs per chromosomes among 7 advanced thymoma patients treated by Pemetrexed.

Chr	Gain			Loss		
	p Arm (%)	q Arm (%)	Whole Arms (%)	p Arm (%)	q Arm (%)	Whole Arms (%)
1	0 (0)	4 (57.1)	0 (0)	0 (0)	0 (0)	0 (0)
2	0 (0)	0 (0)	0 (0)	1 (14.3)	0 (0)	0 (0)
3	0 (0)	1 (14.3)	0 (0)	1 (14.3)	0 (0)	0 (0)
4	0 (0)	0 (0)	0 (0)	0 (0)	0 (0)	0 (0)
5	3 (42.9)	1 (14.3)	0 (0)	0 (0)	0 (0)	0 (0)
6	0 (0)	0 (0)	0 (0)	2 (28.6)	2 (28.6)	0 (0)
7	2 (28.6)	2 (28.6)	0 (0)	0 (0)	0 (0)	0 (0)
8	0 (0)	0 (0)	0 (0)	0 (0)	0 (0)	0 (0)
9	1 (14.3)	0 (0)	0 (0)	0 (0)	0 (0)	0 (0)
10	0 (0)	0 (0)	0 (0)	0 (0)	0 (0)	0 (0)
11	0 (0)	0 (0)	0 (0)	0 (0)	0 (0)	0 (0)
12	0 (0)	0 (0)	0 (0)	0 (0)	0 (0)	0 (0)
13	0 (0)	0 (0)	0 (0)	0 (0)	0 (0)	1 (14.3)
14	0 (0)	0 (0)	1 (14.3)	0 (0)	0 (0)	0 (0)
15	0 (0)	0 (0)	0 (0)	0 (0)	0 (0)	0 (0)
16	0 (0)	0 (0)	0 (0)	0 (0)	0 (0)	0 (0)
17	0 (0)	0 (0)	0 (0)	0 (0)	0 (0)	0 (0)
18	0 (0)	0 (0)	0 (0)	0 (0)	0 (0)	0 (0)
19	0 (0)	0 (0)	0 (0)	0 (0)	0 (0)	0 (0)
20	0 (0)	0 (0)	0 (0)	0 (0)	0 (0)	0 (0)
21	0 (0)	0 (0)	0 (0)	0 (0)	0 (0)	0 (0)
22	0 (0)	0 (0)	0 (0)	0 (0)	0 (0)	0 (0)

Abbreviation: CNVs = copy number variations, Chr = chromosome.

## Data Availability

The datasets presented in this article are not readily available due to ethical concerns regarding patient privacy. Requests to access the datasets should be directed to the corresponding author.

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
