# Peer review of "Whole Exome Sequencing of Thymoma Patients Exhibiting Exceptional Responses to Pemetrexed Monotherapy"

_cancers, 2023, doi:10.3390/cancers15164018_

Round 1
Reviewer 1 Report (Previous Reviewer 2)
Thanks to the extensive work the ms has been improved.
No further comments or rquests.
Reviewer 2 Report (Previous Reviewer 1)
The authors answered and explained all my questions so that I recommend to accept in current format.
This manuscript is a resubmission of an earlier submission. The following is a list of the peer review reports and author responses from that submission.
Round 1
Reviewer 1 Report
The manuscript describes a study conducted by Tomohiro Tanaka and the team to investigate the genetic basis of exceptional responses to Pemetrexed monotherapy in Thymoma patients. Thymoma is a rare type of cancer that affects the thymus gland, which is located in the chest. It is usually treated with surgery, radiation therapy, and chemotherapy.
Pemetrexed is a chemotherapy drug that works by inhibiting the production of DNA and RNA in cancer cells, thereby preventing their growth and division. While Pemetrexed is commonly used for the chemotherapy of advanced Thymoma, exceptional responses to this drug are not frequently observed.
To explore the specific genomic aberrations that lead to an extreme and durable response to Pemetrexed monotherapy, the authors performed whole-exome sequencing on nine formalin-fixed paraffin-embedded tissues from patients with advanced Thymomas treated with Pemetrexed. Among these patients, three were exceptional responders and nine were typical responders.
The study found no differences between the exceptional and typical responders at the level of somatic single-nucleotide variants or copy number variations (CNVs). However, the highest number of arm-level CNVs was observed in an exceptional responder to Pemetrexed. This suggests that arm-level CNVs may play a role in determining the response of Thymoma patients to Pemetrexed monotherapy.
Overall, this study provides additional genomic findings on Thymomas and chemosensitivity. The genetic insights gained from this study may inform future treatments for Thymoma patients who exhibit exceptional responses to Pemetrexed monotherapy. The manuscript is well-written and organized.
In particular, the authors discussed the limitations of the study, which is my major concern in this study.
1. In the conclusion, the author said, “Our study showed that none of the genomic alternations evaluated was specific to exceptional responders to pemetrexed treatment, and no new driver gene was identified” should have a specific condition, e.g “in # samples”.
2. A GO/KEGG analysis of the CNV related genes is suggested to better understand the functional implications of the exceptional responses.
3. Similarly, GO/KEGG analysis of 284 somatic single-nucleotide variants would provide us with some functional implications.
Reviewer 2 Report
The authors discuss genomic data in relation to effectiveness of therapy in Thymomas.
Please note that the name of genes are usually written in italics; change
Line 23 arm level: specify the meaning also in the abstract
Line 30-31: quoting the total number of thymoma patients is misleading, as molecular analyses were performed in a smaller number of cases; consider erasing from “Among” to “responders”.
Line 36: specify the type of differences not found e.g. in number of variants , size of CNV and so on.
Line 41: erase “based”
Line 42-43: “exceptional response; extraordinary response;” the two types of response are not differentiated in the text, use only one.
Line 82: change “m2” in “m2”
Line 88- 91: the definition of “ exceptional response” in thymomas is a new one formulated by the authors or refers to published data? If already reported, a reference should be cited here
Fig. 1 please consider including also data from WHO classification
Line 156: “The median PFS of exceptional responders was longer than that of typical responders (22.5 vs. 157 3.1 months; p = 0.37) (Figure 3A–B)”.
If exceptional responders are first defined as patients with better outcome, it is obvious that they perform better.
Line 180: “Furthermore, 17 SNVs were identified in genes predicted to be potentially pathogenic by at least one method.”
I guess the authors mean that the variants (not genes) are potentially pathogenic; they should add to the table the prediction tool they used; the functional significance of the genes containing the variants should be discussed even briefly, in the appropriate section.
Line 199 to 204
“p” and “q” are longstanding international cytogenetic terms indicating , respectively; so it is not needed to repeat “arms” ; for instance “5p” is the usual cytogenetic way of reporting, not “5p arm” . In addition, chromosomes 13 and 14 are acrocentric chromosomes, whole arm duplication refers to tandem duplication or isochromosome? Whole arm deletion implys that only the centromeric region is conserved?
Line 209 and Fig. 5: “whereas cluster 3 (patients 6 and 9) showed arm-level chromosome gain” but in Fig. 5 pt 6 seems to have mainly losses (red bars)
Line 218: “We identified certain SNVs and Indels that were potentially deleterious, including SCN3A, GTF3C1, and RPL5” SNVs do not include genes; modify.
Line 219: “However, none of the genomic alternations (missprint, modify) were specific to exceptional responders. Interestingly, arm-level chromosomal CNVs were detected in a thymoma patient who exhibited an exceptional response to pemetrexed.”
But the so called arm levels CNV were found also in “non exceptional” cases; modify the sentence.
Line 254: “In particular, an exceptional responder (patient 8) had a whole-arm gain and loss of chromosome 3 that contained the CAND2 mutation, which may be related to hypersensitivity to pemetrexed treatment “
Poorly written from a genetic point of view: i) gain and losses are not the same from a genetic functional point of view; the gain and losses may contain the CAND2 gene, NOT the CAND2 mutation, unless specific mutation was identified.
The mechanism by which CAND2 may be related to pemetrexed should be discussed and references added.
Line 262: “In particular, only two exceptional responders were included, one of whom was excluded from the analysis due to a small amount of genomic data. “
So in fact data are available just for one single case!!
Line 276 to the end: the reported conclusions cannot be obtained on a single case.
Line 194
Definition of arm level CNV seem different from the usual ; see: Ondrej Pös, et al. DNA copy number variation: Main characteristics, evolutionary significance, and pathological aspects,
Biomedical Journal, Volume 44, Issue 5, 2021, Pages 548-559, ISSN 2319-4170, https://doi.org/10.1016/j.bj.2021.02.003.
“Cancer
Although cancer may be monogenic or, more typically, multifactorial, oncogenetics tends to be considered as a special field and discussed separately. In cancer genetics, CNVs are divided into two classes based on their size: i) large-scale, also known as chromosome-arm level variants encompassing >25% [115] or â…“ of the chromosome arm [116]”
Quality of English Language is good; just a few improvement can be made